# Immune Checkpoint Inhibitors in HBV-Caused Hepatocellular Carcinoma Therapy

**DOI:** 10.3390/vaccines11030614

**Published:** 2023-03-08

**Authors:** Jin Zhang, Changwei Hu, Xiaoxiao Xie, Linzhi Qi, Chuanzhou Li, Shangze Li

**Affiliations:** 1School of Medicine, Chongqing University, Chongqing 400044, China; 2Department of Medical Genetics, School of Basic Medicine, Tongji Medical College, Huazhong University of Science and Technology, Wuhan 430030, China

**Keywords:** hepatocellular carcinoma, HBV, immunotherapies, immune checkpoint inhibitors

## Abstract

Hepatitis B virus (HBV) infection is the main risk factor for the development of hepatocellular carcinoma (HCC), the most common type of liver cancer, with high incidence and mortality worldwide. Surgery, liver transplantation, and ablation therapies have been used to treat early HBV-caused HCC (HBV-HCC); meanwhile, in the advanced stage, chemoradiotherapy and drug-targeted therapy are regularly considered, but with limited efficacy. Recently, immunotherapies, such as tumor vaccine therapy, adoptive cell transfer therapy, and immune checkpoint inhibitor therapy, have demonstrated promising efficacy in cancer treatment. In particular, immune checkpoint inhibitors can successfully prevent tumors from achieving immune escape and promote an anti-tumor response, thereby boosting the therapeutic effect in HBV-HCC. However, the advantages of immune checkpoint inhibitors in the treatment of HBV-HCC remain to be exploited. Here, we describe the basic characteristics and development of HBV-HCC and introduce current treatment strategies for HBV-HCC. Of note, we review the principles of immune checkpoint molecules, such as programmed cell death protein 1(PD-1) and cytotoxic T-lymphocyte-associated protein 4 (CTLA-4) in HBV-HCC, as well as related inhibitors being considered in the clinic. We also discuss the benefits of immune checkpoint inhibitors in the treatment of HBV-HCC and the efficacy of those inhibitors in HCC with various etiologies, aiming to provide insights into the use of immune checkpoint inhibitors for the treatment of HBV-HCC.

## 1. Introduction

According to the Global Cancer Statistics 2020, primary liver cancer is the most prevalent and one of the three major cancers, with approximately 906,000 new cases and 860,000 deaths annually; these numbers from liver cancer, predictably, will increase by over 55% by 2040 [1,2]. Its occurrence is geographically diverse, with the majority of cases occurring in Eastern Asia and Northern Africa, and the incidence and mortality rate in men are significantly higher than those in women. Regarding the lesion location, primary liver cancer can be classified as intrahepatic cholangiocarcinoma or hepatocellular carcinoma (HCC).

HCC is the most common type of liver cancer, accounting for 75% to 85% of cases [3], and can be loosely classified into viral and non-viral varieties, depending on whether it is induced by a virus. HCC caused by the hepatitis B virus (HBV-HCC) and HCC caused by the hepatitis C virus (HCV-HCC) are two common examples of viral HCC, while non-viral HCC is primarily caused by alcoholism, obesity, and smoking, with nonalcoholic steatohepatitis and diabetes mellitus also being potential causes, and non-alcoholic fatty liver disease (NAFLD)-HCC being the most common form of non-viral HCC [4,5,6].

As a liver metabolic disorder, NAFLD has been attributed to risk factors including obesity, type 2 diabetes, hyperlipidemia, etc. NFALD is mainly prevalent in the Middle East and South America, affects more than 100 million adults [7], and its overall pooled HCC incidence rate among NAFLD patients is 1.25 per 1000 person-years [8]. Despite this, in contrast to NAFLD, HBV and HCV lead to HCC mainly through blood transmission, mother–baby transmission, and sex transmission, which are more prone to spreading. HCV is a positive RNA virus that cannot be completely cleared by the immune system; therefore HCV-HCC is always manifested by chronic inflammatory stimulation [9]. Statistically, HCV is more prevalent in the Eastern Mediterranean region, European region, and South-East Asia, with an estimated 71.1 million individuals infected worldwide; as a result, HCV infection is responsible for almost 20% of HCC patients [2,10]. Other than HCV, infection by HBV with partially double-stranded DNA not only results in inflammatory stimulation, but also integrates the viral DNA into the host DNA [11]. As the leading cause of liver cancer, HBV is mainly spreading in Africa and Eastern Asia, especially in China and Mongolia. As reported, HBV-related diseases, in 2019, resulted in 555,000 deaths, accounting for 48.8% of all hepatitis-related deaths [12]. In general, although vaccination, to some extent, has prevented HBV infection, the burden of HBV and HCC induced by HBV is still enormous; therefore, it is urgent to seek effective treatment strategies for HBV-HCC.

Early HBV-HCC can be treated with traditional therapies, such as surgery, chemotherapy, and liver transplantation. However, for the reasons that HCC is occult and it’s difficult to diagnose at an early stage, the majority of clinical HCC patients are at an advanced stage. Additionally, the treatment options for the advanced stage are extremely limited and have little efficacy, which ultimately results in poor prognosis. As alternative therapeutic possibilities, tyrosine kinase inhibitor therapy, chimeric antigen receptor T-cell immunotherapy (CAR-T), and immune checkpoint inhibitor therapy have been developed in recent years. Immune checkpoint inhibitors enhance the anti-tumor immune response by activating the killing ability of immune cells, especially T lymphocytes, through external intervention, which is important for the treatment of advanced HCC. In this review, we demonstrate the molecular characteristics of HBV-HCC and the relationship between HBV genotypes and HCC, discuss the mechanisms of HBV infection promoting the occurrence and development of HCC, and review the treatment status of HBV-HCC. Furthermore, we introduce the known and potential immune checkpoints identified to date and their principles and corresponding inhibitors. Additionally, we review the advantages of immune checkpoint inhibitors in the treatment of HBV-HCC and their distinct efficacies in dealing with varying etiologies caused by HBV.

## 2. Molecular Characterization and Typing of HBV-HCC

### 2.1. Molecular Characterization of HBV-HCC

Patients with HBV-HCC exhibit altered gene expression, as well as particular genetic mutation profiles. Genes linked to oxidative stress and T cell immunological responses are significantly downregulated in HBV-HCC, and these patients express higher levels of miR-143, miR-34, and miR-19, rendering HBV-HCC more aggressive [13]. Additionally, it has been discovered that HBV-HCC is linked to innate immune response pathways, primarily including acute-phase response signals, complement system signals, and primary immunodeficiency signals, manifested by the elevation of innate immune response-related genes, such as MIF, NOS2, and TALDO1 [14].

Variations in the molecular features of HBV-HCC and HCV-HCC have been identified. HCV-HCC is characterized by the enrichment of downregulated genes linked to T cell activation, oxidative stress, the activation of M2-like CD68+ cells, and the considerable downregulation of immune cell infiltration, whereas HBV-HCC presents significant upregulation of immune cell infiltration [14]. NFE2L2, a crucial transcription factor involved in oxidative stress responses, is often mutated in non-HBV HCC, but not in HBV-infected tumors when comparing the somatic mutation profiles of HBV and non-HBV HCC. This suggests that, in HBV-HCC, the frequency of oxidative stress responses activated by gene mutations is lower [15]. Additionally, a direct relationship has been demonstrated between the etiology and the activation mechanism of the Wnt/β-catenin signaling pathway in HCC [16]. Through the activation of CTNNB1 mutations or the inactivation of AXIN1/APC mutations, Wnt/β-catenin pathway activation contributes to the development of HCC. Compared with alcoholic, HCV, or non-alcoholic steatohepatitis (NASH), HBV-HCC has a much lower prevalence of β-catenin mutational activation. Tumor suppressor gene, TP53, mutations are also strongly linked with shorter disease-specific survival and worse prognosis in HBV-HCC, although TP53 mutations are uncommon in non-HBV tumors [17].

### 2.2. Typing of HBV-HCC

Currently, there are nine recognized genotypes of HBV (A–I) and a putative 10th genotype, “J” [18]. In clinical practice, genotypes are closely associated with HCC’s occurrence, and various genotypes have different prognoses and risks. It has been demonstrated that genotype F shows the greatest HCC incidence, followed by genotypes C and A, all of which have much higher rates than genotypes D and B [19]. Additionally, genotype C is consistently more prevalent in HBV-HCC, and it has been found that the HCC risk of genotypes D and C is more than eight times higher compared with genotype B and co-infected genotype BC [20]. The genome sequences of genotypes A–H have been extensively analyzed, and the results show that, among the related mutations of immune escape, genotype B has the highest escape mutation rate, while genotype H has no mutation. Further, in antiviral drug resistance and HCC development, genotype G mutations are more common than those of genotype C, which has a higher prevalence of chronic HCC development than genotypes A and B [21,22]. However, the DNA sequences of 426 HBV-HCC patients reveal that integration into the host DNA by genotype B is more obvious than that by genotype C [23].

## 3. Occurrence and Development of HBV-HCC

### 3.1. Direct Oncogenic Mechanism of HBV

HCC is linked to persistent HBV infection in direct and indirect ways, and HBV DNA could integrate into host genes and lead to retroviral insertion mutations [24]. This finding supports the notion that HBV can not only cause liver cirrhosis by chronic inflammation, but also promotes malignant transformation by direct carcinogenesis. There are three direct oncogenic pathways of HBV: insertion mutation, the facilitation of genomic instability, and the carcinogenesis of viral proteins (Table 1).

#### 3.1.1. Insertion Mutations

Statistically, during the early stage, approximately 85% of HBV-HCC tissue samples display an incidence of HBV DNA integration [25]. The genes X, C, S, and the enhancer of HBV are integrated into the host genome, with the X and C genes constituting the majority of integration sequences. The integration of the X gene directly activates oncogenes and decreases the expression of tumor suppressor genes [26]. It has been discovered that HBV integration regularly affects 15 genes, which are strongly related to HBV-HCC. Moreover, integration into the TERT, CCNA2, CCNE1, and KMT2B genes is more common in the HCC group than in the non-tumor group infected with HBV. The TERT gene contributes to the formation of telomerase, an enzyme that maintains telomeres and protects chromosomes, and HBV DNA has been found to repeatedly integrate into this gene location, promoting chromosomal abnormalities [27]. The insertion of HBV DNA into the CCNA2 and CCNE1 genes, which are associated with the G1/S transition of the cell cycle, disrupts the biological process and promotes tumor development; KMT2B is a tumor suppressor gene that prevents the uncontrolled growth and division of cells. The integration of HBV DNA into the KMT2B gene can promote the infinite proliferation of liver cancer cells. In addition to these genes, HBV integration has also recently been found to impact genes, including PTPRD, UNC5D, and NRG3, for which the transcription and protein levels are influenced [23]. Previous studies have evaluated 18,596 HBV integration sites and revealed a total of 396 frequently targeted genes (RTGs), with TERT and KMT2B being the two most prevalent hits [25]. In summary, HBV DNA integration will cause the accumulation of mutations and eventually promote the incidence of HCC.

#### 3.1.2. HBV Causes Genomic Instability

HBV preferentially integrates into chromosomal duplication or vulnerable regions during tumorigenesis, and integration sites are significantly enriched in telomeres, indicating that it has a propensity to integrate into components that maintain chromosomal stability [23]. Whole-genome sequencing of HBV-HCC has revealed a large increase in copy number variation (CNV) at HBV breakpoints that may cause chromosomal instability [28]. Additionally, the interference of mitotic checkpoints by HBV x protein (HBx), which interacts with the DNA damage-binding protein 1(DDB1), causes chromosomal instability and facilitates the development of HCC [29].

#### 3.1.3. Prolonged Expression of Viral Proteins Affects Cell Functioning

By interfering with cell functioning, HBV viral proteins, such as HBx, HBc, and preS, could promote HBV carcinogenesis. In HBV-replicating cells, HBx is recruited onto covalently closed circular DNA (cccDNA) small chromosomes to increase transcription and sensitize hepatocytes to oncogenesis, which contributes to the carcinogenicity of HBV. Furthermore, HBx regulates cccDNA transcription/viral replication [30], and also controls the expression of the host and HBV genomes, which leads to a number of significant epigenetic alterations, such as DNA methylation, histone modification, chromatin remodeling, and microRNA dysregulation, thus resulting in the occurrence and progression of HCC [31]. The hepatitis B capsid protein, known as HBc, has been identified to promote chronic infection in HBV and hepatocyte death by blocking death receptor 5 (DR5) [32]. Moreover, it has been demonstrated that HBc could suppress host antiviral immune responses by interacting with the promoters of INF-induced genes [33]. PreS2 deletion also increases the risk of HCC by modifying inflammatory and metabolic pathways. PreS mutations in HBV cause decreased secretion of viral proteins and particles, promote the accumulation of HBsAg in the endoplasmic reticulum, and induce endoplasmic reticulum stress to stimulate the occurrence of HCC [34,35].

### 3.2. Immune-Mediated Indirect Oncogenic Mechanisms of HBV

The immune response contributes significantly to the progression of chronic hepatitis B to HCC [36]. HBV infection disrupts the HBV-HCC tumor immune microenvironment and promotes the development of HCC. Various immune cells, including T cells, B cells, and NK cells, produce significant responses to control and eliminate HBV in the early stages of infection [37]. When acute HBV infection is left untreated, it turns into chronic HBV infection, and adaptive immunity shifts from immunological tolerance to progressive immune activation, inactivation, reactivation, and fatigue. This immune imbalance indirectly accelerates the incidence and progression of HCC [38]. Large numbers of virus-specific lymphocytes, such as HBV-specific CD8+ T cells, infiltrate hepatitis lesions and are readily identifiable. These cells secrete cytokines to eliminate infected host cells, but, due to the persistently high content of HBV DNA, immune responses are suppressed, with the appearance of highly expressed inhibitory receptors, such as cytotoxic T-lymphocyte-associated protein 4 (CTLA-4) and programmed cell death protein 1 (PD-1) [39]. The dysfunction of depleted HBV-specific CD8+ T cells prevents HBV from being fully eliminated, causing persistent inflammation and fostering hepatocellular carcinogenesis. Furthermore, CD4+ T cells also play roles in the tumor immune microenvironment. For instance, during the development of HCC, regulatory T cells (Treg cells) produce cytokines such as IL-2, IL-10, TGF-β, and IL-35, which restrain the immune response to the HBV antigen and HCC tumor antigen [40]. Besides this, in persistent HBV infection, NK cells are inhibited because of their antiviral activity, the expression of NK cell inhibitory receptors TIM-3 and PD-1 is upregulated, while the secretion of IFN-γ and TNF-α is decreased, and the secretion of IL-10 and TGF-β, inhibiting NK cell maturation, is increased. These factors lead to the inability of NK cells to clear the HBV virus and also mediate HCC [38]. Furthermore, it has been shown that persistent HBV infection can accelerate the development of HCC by causing the dysregulation of the complement system and a cytokine imbalance [41,42,43]. Taken together, HBV infection suppresses immune system activation and leads to the formation of an inhibitory microenvironment, which is a major contributor to the development of HCC.

## 4. Current Status of Treatment of HBV-HCC

The most up to date treatment strategies for HBV-HCC are summarized in below (Figure 1).

### 4.1. Antiviral Therapy

The immediate goal of antiviral therapy is to reduce the amount of HBV in patients, protecting the body from being damaged; after this, other treatment strategies for HCC can be applied. The most widely used antiviral treatment drugs include cytokines, nucleotide analogs (NAs), and small interfering ribonucleic acids (siRNAs). Cytokine IFN-α has the potential to cure HBV infection by eliminating cccDNA in patients, but its effectiveness in clinical settings still needs to be improved. NAs, such as lamivudine and entecavir, can directly target the viral protein and DNA to inhibit HBV; however, they cannot function well to eliminate cccDNA. Finally, siRNAs, with specific nanoparticles, have the capacity to degrade mRNA to prevent the synthesis of HBV proteins [44,45,46]. Previous studies have demonstrated that antiviral therapy could significantly decrease the damage caused by tumors in patients with HBV-HCC and increase survival rates [47,48].

### 4.2. Surgical Excision and Radiofrequency Therapy

Surgical excision and radiofrequency ablation are the primary options for the treatment of early HCC patients. Surgical resection, by moving tumor tissue from the liver, increases patient survival rates and has always been considered. However, it is likely that there will also be severe postoperative problems, such as recurrence or tumor cell diffusion [49]. The principle of radiofrequency ablation is to destroy tumor tissue using high temperatures. Although this technique is highly effective and safe for the treatment of HCC, it is difficult to control the treatment area and it may increase the risk of damaging healthy tissue; furthermore, the heat-dissipation effect of this approach may also facilitate the diffusion of tumor cells [50].

### 4.3. Liver Transplantation

For early HCC patients meeting the “Milan criteria” (single cancer of liver ≤ 5 cm or three nodules ≤ 3 cm) and without spreading or large blood vessel invasion, liver transplantation could offer an effective treatment strategy that can significantly increase patient survival rates and reduce the likelihood of recurrence [51,52]. Furthermore, alphafetoprotein (AFP) could be a useful biomarker for predicting the outcome of liver transplantation; the lower the amount of AFP, the better the treatment effect [53]. Nevertheless, liver transplantation still faces some limitations, including the challenge of finding suitable donors and the probability of liver bleeding after transplantation; therefore, more effective therapeutic strategies are urgently needed [54,55].

### 4.4. Radiotherapy and Chemotherapy

In order to eliminate HCC, transarterial radioembolization is employed to release large doses of ionizing radiation elements to penetrate the tumor through the hepatic artery. By focusing a high dosage of radioactive elements on the tumor, the risk of radioactive elements damaging healthy liver tissue is relatively reduced [56]. However, remaining radioactive materials after tumor therapy might harm the healthy liver tissue and cells, leading to poor prognosis [57]. Meanwhile, local chemotherapy is a critical strategy in the treatment of HCC. In particular, transarterial chemoembolization (TACE) and hepatic arterial perfusion chemotherapy (HAIC) involve a variety of techniques to deliver chemical drugs to the area around the tumor cells and prevent these cells from undergoing DNA replication and transcription [58].

### 4.5. Tyrosine Kinase Inhibitor Therapy

Given that the abnormal activation of phosphate groups caused by tyrosine kinases is closely related to the occurrence and development of tumors, including HCC, the competitive role of tyrosine kinase inhibitors (TKIs) in kinase receptors is crucial for the treatment of tumors [59]. TKIs, such as Sorafenib and Lenvatinib, could effectively decrease the activity of tyrosine kinases, inhibiting the oncogenic signal pathway and thus achieving the treatment of advanced HCC [60]. Moreover, in clinical practice, TKI therapy combined with other treatment strategies, such as immunotherapy, could be an effective therapeutic option. However, adverse reactions to TKIs are also common; for instance, disorders of the digestive tract, skin, and other organs may appear after treatment.

### 4.6. Immunotherapy

The immune system serves as the main defense against external threats, but its dysfunction also conversely contributes to the initiation and growth of tumors. CAR-T, tumor vaccines, and immune checkpoint inhibitor therapy are the three primary anti-tumor therapies based on the immune system; they mainly entail the surveillance and killing of T lymphocytes and antibody production capacity of B lymphocytes. Tumor vaccines target antigens generated by tumor cells and increase the potency of tumor-specific responses; CAR-T therapy is used to treat tumors by transferring modified T cells into the patient to enhance their immune function against tumor cells; and inhibiting the activity of immunological checkpoints aims to prevent the negative regulation of the immune response, enhance the capacity of immune cells to eradicate tumor cells, and finally control the formation and growth of malignancies [61,62,63].

## 5. Immune Checkpoint Inhibitors for the Treatment of HBV-HCC

### 5.1. Principle of Action of Immune Checkpoint Molecules in HBV-HCC and Targeted Inhibitors

In the tumor immune microenvironment, T cells are able to recognize tumor antigens presented by antigen-presenting cells, and activated T cells negatively regulate immune responses through immune checkpoints. T cell activation by antigen-presenting cells requires two signals: the interaction between T cell membrane surface receptors and MHC molecules on the antigen-presenting cell surface, and the interaction between T cell co-receptor signals [64]. The T cell surface expresses two types of co-receptor signals: one is a co-stimulatory receptor that can activate T cells positively; the other is a co-suppressor receptor that suppresses T cells [64,65]. Immune checkpoint inhibitor therapy mainly aims to reverse the exhaustion state of CD8+ T cells or NK cells based on co-receptor signaling, reducing the function of co-suppressor receptors.

In recent years, a series of immune checkpoints have been discovered (Figure 2). Under normal circumstances, immune checkpoints regulate immune responses to maintain immune tolerance; when tumors invade, immune checkpoints are abnormally activated to form a tumor immunosuppressive microenvironment, which aids tumor cells in evading immune attack and promoting the growth and migration of hepatoma cells. With the deepening of research on immune checkpoints, inhibitors have gradually been developed and applied, and obvious advantages and prospects in liver cancer have been shown, providing new treatment strategies for advanced HCC. Many immune checkpoints have been identified and studied in HBV-HCC, including programmed death protein 1 (PD-1), cytotoxic T lymphocyte-associated protein 4 (CTLA-4), and recently discovered immune checkpoints, such as lymphocyte activation gene 3 (LAG-3), T cell immunoglobulin and ITIM domain (TIGIT), T cell immunoglobulin mucin 3 (TIM-3), etc. Immune checkpoints, as membrane proteins, are initially expressed in the endoplasmic reticulum and are transported onto the cell’s surface, and glycosylation ensures that only mature and functional immune checkpoints are delivered to the cell’s surface [66]. On the cell’s surface, immune checkpoints regulate the surface protein levels through internalization, circulation, and ubiquitination, mediating cell signaling. The immune surveillance function of immune cells in the HCC microenvironment is often blocked through multiple mechanisms, and signal suppression is a major cause of immunosuppression [67]. Hepatoma cells downregulate the activity of stimulatory immune receptors and upregulate the activity of inhibitory immune receptors, thereby causing immune evasion and immunosuppression. Taking T lymphocytes as an example, hepatoma cells can not only decrease the expression of MHC-I on the surface, but also regulate inhibitory signaling pathways by inducing the levels of immunosuppressive receptors on the cell surface, such as CTLA-4 and PD-1. The principles of HCC immune checkpoints in HBV-HCC and their corresponding inhibitors are illustrated below. The clinical trials of immune checkpoint inhibitors in HBV-HCC are also listed (Table 2).

#### 5.1.1. Programmed Cell Death Protein 1 (PD-1)

PD-1 is a type I transmembrane protein belonging to the CD28 receptor family with two ligands, PD-L1 and PD-L2, of which PD-L2 has three-times-higher affinity to the receptor than PD-L1 [74]. PD-1 is mainly expressed in memory CD8+ T cells, memory CD4+ T cells, tumor cells, antigen-presenting cells, and endothelial cells [75]. PD-L1 and PD-L2 are mainly expressed on B cells, macrophages, monocytes, and dendritic cells [76]. The intracellular domain of PD-1 is composed of immune receptor tyrosine switch motifs (ITSM) and immune receptor tyrosine inhibition motifs (ITIM). When PD-L1/PD-L2 binds to PD-1, ITIM and ITSM are phosphorylated, and a large number of tyrosine phosphatases, mainly SHP-2, are recruited in the PD-1 cytoplasmic tail. Then, SHP-2 downregulates co-stimulatory signals via the dephosphorylation of key components of TCR, and serves an inhibitory function in downstream signaling molecules, including CD28, ZAP70, PLCγ, and PI3K, thus mediating T cell failure and inhibiting anti-tumor cell immunity [77,78]. Therefore, the PD-L1 expressed by tumor cells inhibits the proliferation and activation of T cells by binding to PD-1 on T cells, so tumor cells obtain immune tolerance [79]. Moreover, the higher the expression level of PD-L1 in liver cancer tissue, the higher the degree of tumor malignancy, the stronger the aggressiveness, the higher the likelihood of relapse after surgery, and the shorter the overall survival of patients [80].

At present, the US Food and Drug Administration (FDA) has approved the marketing of four PD-1 inhibitors and three PD-L1 inhibitors. In 2014, as the first PD-1 inhibitor approved by the FDA, pembrolizumab showed significant efficacy in a number of tumor treatments. On 9 November 2018, the FDA accelerated the approval of pembrolizumab for patients with HCC treated with sorafenib. Nibolumb was the second PD-1 monoclonal antibody approved in 2015. Based on the results of the Checkmate-040 study, the FDA approved nivolumab for patients with advanced HCC who have been treated with sorafenib. After nivolumab was approved for the treatment of advanced HCC, a multicenter retrospective study evaluating the efficacy and safety of nivolumab monotherapy in the treatment of advanced HCC showed that nivolumab had an objective response rate (ORR) of 22.4 percent and a disease control rate (DCR) of 52.1 percent, which are comparable to the results of Checkmate-040, revealing the good safety profile of nivolumab for advanced HCC [81]. In addition, cemiplimab was approved by the FDA in 2018. A phase 2 trial at the Icahn School of Medicine at Mount Sinai evaluated cemiplimab in patients with resectable HCC. The death of large populations (≥50%) of tumor cells was induced by cemiplimab before surgery in nearly one-third of patients. This is, to date, the largest clinical trial in the field of neoadjuvant PD-1-targeted monotherapy for liver cancer [82]. Cemiplimab, however, is currently only licensed for three indications and cannot be used to treat HCC. In 2021, the FDA approved Dostarlimab, the fourth PD-1 monoclonal antibody, and on 21 October 2022, the FDA approved Dostarlimab in combination with tremelimumab for the treatment of unresectable HCC.

#### 5.1.2. Cytotoxic T-Lymphocyte-Associated Protein 4 (CTLA-4)

CTLA-4 is expressed on the surfaces of activated immune cells, including T cells, B cells, and NK cells [83]. It is a transmembrane protein belonging to the immunoglobulin family, the extracellular domain of which binds to the B7-1 (CD80) and B7-2 (CD86) ligands. The tyrosine in the YVKM motif of the cytosolic domain of CTLA-4 can be phosphorylated by kinases, such as Src family kinases, and phosphorylated YVKM recruiting SHP2 and PP2A inhibits AKT activity and prevents the interaction between CTLA-4 and AP-2, thereby maintaining CTLA-4’s transmission of inhibitory signals on the cell surface [84]. Compared with CD28, CTLA-4 has a higher affinity for CD80/86, and the difference in affinity leads to the reduced activation of T cells through ligand competition with CD28, thereby promoting tumorigenesis [85]. In addition, CTLA-4 also induces the expression of diamide 2,3-dioxygenase (IDO) in dendritic cells by linking CD80/86, resulting in tryptophan depletion and T cell inhibition [86].

Currently, the FDA has approved a total of two CTLA4 inhibitors. Ipilimumab was the first CTLA4 monoclonal antibody approved by the FDA in 2011, and on 10 March 2020, the FDA accelerated the approval of nivolumab and ipilimumab in combination for HCC treated with sorafenib as the first dual immunotherapy approved for the patient population with advanced HCC. According to the checkMate-040 Phase I/II study’s findings in the nivolumab monoclonal antibody cohort, nivolumab with ipilimumab has an acceptable safety profile and a long-lasting effect on advanced HCC [87]. Tremelimumab is the second FDA-approved CTLA-4 monoclonal antibody. HIMALAYA phase III trial data show that tremelimumab plus durvalumab can significantly improve overall survival (OS) compared with sorafenib in first-line patients with unresected HCC, regardless of baseline albumin–bilirubin (ALBI) grade. On 21 October 2022, tremelimumab in combination with durvalumab was approved by the FDA for the treatment of unresectable HCC [88].

#### 5.1.3. Lymphocyte Activation Gene 3 (LAG3)

LAG3 belongs to the type I Ig family of inhibitory receptors and is expressed on activated T, B, and NK cells [89]. Resting CD8+ T cells express LAG3 at low levels, and the LAG3 levels are substantially upregulated when antigens are present [90]. LAG3, sharing approximately 20% of highly conserved structural motifs identical to CD4, is a ligand for MHC-II class molecules; as the interaction of LAG3 with MHC-II involves amino acid residues in the proline-rich D1 rings, LAG3 has a higher affinity than CD4 and it inhibits the activation of CD4+ T cells by competitively preventing CD4 from interacting with MHC-II [91]. When CD3 and LAG-3 are crosslinked, inhibitory signals are activated to prevent T cell proliferation, cytokine production, and calcium flux [92]. In addition, LAG3 is believed to be associated with T cell homeostasis, chronic viral infections, parasitic infections, and cancer [93,94]. When TCR is involved, the LAG3 cytoplasmic tail mediates inhibitory signals through three conserved motifs, namely a KIEELE motif, a glutamate proline dipeptide multi-repeat motif (EP motif), and a serine phosphorylation motif (S484). A lysine residue at 468 of the KIEELE motif is required for the downstream inhibition of signaling that prevents T cell proliferation, and the absence of this region prevents negative signals of T cell function [95]. In addition, recent studies have found that LAG3 binds to the TCR-CD3 complex and inhibits signaling by driving co-acceptor–Lck dissociation [92]. In fact, the inhibitory function of LAG-3 in CD8+ T cells does not involve MHC-II. In the immune microenvironment of HCC, LAG-3 can inhibit T cell function by binding to ligands, such as alpha-synaptic nucleoprotein fibrils (αSYN), liver sinusoidal endothelial cell lectin (LSECtin), galectin-3 (Gal-3), or fibrinogen original protein 1 (FGL-1), which are secreted by HCC cells [96]. Anti-FGL-1 antibody and anti-LAG-3 therapy inhibit the proliferation of liver cancer cells, reduce the size of tumors, and delay the progression of liver cancer [97].

Relatlimab is the first LAG-3 inhibitor and the third immune checkpoint after PD-1 and CTLA-4. On 18 March 2022, the FDA approved a fixed-dose combination of relatlimab with nivolumab for the treatment of adult and pediatric patients 12 years of age or older with unresectable or metastatic melanoma, but it is not currently approved for HCC. Several clinical trials with relatlimab for the treatment of HCC are currently ongoing, including an international multicenter (including China) phase 1/2 clinical trial of LAG-3 antibody triple therapy. This was initiated to evaluate the initial efficacy and safety of relatlimab in the first-line treatment of advanced/metastatic HCC in combination with the PD-1 inhibitor nivolumab and the VEGF inhibitor Bevacizumab (NCT05337137).

#### 5.1.4. T Cell Immunoglobulins and ITIM Domains (TIGIT)

TIGIT consists of an extracellular immunoglobulin variable set (IgV) domain, a type I transmembrane domain, and an intracellular domain containing ITIM and immunoglobulin tyrosine tail (ITT)-like motifs [98]. TIGIT is mainly expressed on T cell subsets and NK cells, and its recognizable ligands have been reported to be CD155, CD112, and CD113; in particular, CD155 has the highest affinity for TIGIT [99]. Most patients with HCC develop the condition following cirrhosis, so the HCC tumor microenvironment is rich in cancer-associated fibroblasts (CAFs), which have been shown to downregulate CD155 on the cell surface to reduce binding to DNAM-1 and block the killing activity of NK cells [100]. In addition, due to the fact that TIGIT has a higher affinity for CD155 than DNAM-1, the ligand competition inhibition prevents the activation of the co-stimulatory molecule DNAX helper molecule-1 (DNAM-1) and transmits inhibitory signals to T cells [101]. In 2009, TIGIT was first identified by Yu et al. as an immune checkpoint that inhibits T cell activation, and the TIGIT/CD155 pathway may exert its immunosuppressive function by inducing the DC secretion of the anti-inflammatory cytokine IL-10 in HCC [98]. In NK cells, TIGIT and CD96 are co-inhibitory receptors, and TIGIT inhibits the PI3K and MAPK signaling pathways by recruiting SHIP1 via growth factor receptor-binding protein 2 (Grb2) and SHIP1 via β-inhibitory protein 2 (β-arrestin2) to inhibit the NF-κB pathway, thereby reducing the secretion of cytokines, such as IFN-γ and TNF-α, and inhibiting anti-tumor immunity [102]. In tumor tissue, TIGIT+NK cells can be transformed into CD96+ NK cells, which further weakens the anti-tumor activity of NK cells [103]. Studies have shown that peripheral blood T lymphocytes and regulatory T cells in HCC patients significantly upregulate TIGIT and CD155 expression, and are associated with poor HCC prognosis [104]. Upregulated TIGIT exhibits immunosuppressive effects by mediating the functional depletion of CD8+ T lymphocytes, which is closely related to the poor prognosis of HBV-HCC [105,106]. In addition, TIGIT+ Treg cells also mediate HCC immune escape by accumulating in the tumor microenvironment and inhibiting the anti-tumor response [107].

Although there are currently no FDA-approved inhibitors of TIGIT, a series of clinical trials including advanced HCC are still ongoing. For example, a phase Ib/II, open-label, multicenter, randomized umbrella study has been initiated to recruit patients with locally advanced or metastatic HCC who have not received systemic therapy in order to evaluate the efficacy and safety of multiple combination therapies based on immunotherapy in patients with advanced HCC. It is worth mentioning that the inhibitors used in this clinical trial include the TIGIT inhibitor Tiragolumab (NCT04524871). In addition, another TIGIT inhibitor, Vibostolimab, is also undergoing clinical trials related to advanced HCC to determine the safety, tolerability, and preliminary efficacy of a pembrolizumab/vibostolimab combination formulation (MK-7684A) with or without other anticancer therapies in selected patients with advanced solid tumors, including HCC (NCT05007106).

#### 5.1.5. T Cell Immunoglobulin Mucin 3 (TIM-3)

TIM-3, also known as hepatitis A virus cell receptor 2 (HAVCR2), is a type 1 transmembrane protein that is expressed on a variety of immune cells, including NK cells, dendritic cells, T cells, and macrophages, and is the most expressed in NK. TIM-3 mainly has four ligands, namely C-type lectin hemagglutinin 9 (Galectin9), high-mobility group protein 1 (HMGB1), carcinoembryonic antigen cell adhesion molecule 1 (Ceacam1), and non-protein ligand phosphatidylserine (PS). An increase in TIIM-3 expression is found in CD41 and CD81 T cells in HCC, and the TIM-3/galectin-9 signaling pathway mediates T cell senescence in HBV-HCC and predicts adverse outcomes in patients with HBV-HCC [108]. In addition, serum-soluble TIM-3 is involved in disease progression and HCC development in chronic HBV infection, and high levels of serum TIM-3 increase the risk of liver cancer in patients with chronic HBV infection and are related to low survival in patients with HBV-associated HCC [109]. Both PD-1 and TIM-3 have been found to be highly expressed in liver cancer tissue and are associated with their gene polymorphisms in patients with HBV-HCC; moreover, increased expression of PD-1 and TIM-3 is also significantly associated with an elevated tumor grade [110].

To date, no FDA-approved inhibitors of TIM-3 are available, but a series of clinical trials, including advanced HCC, are ongoing. For example, TSR-022 (cobolimab, TIM-3-binding antibody) and TSR-042 (dostarlimab, PD-1-binding antibody) may stop the growth of tumor cells by activating the immune system to attack the tumor. A phase II trial is investigating the combination of TSR-022 (cobolimab, TIM3-binding antibody) and TSR-042 (dostarlimab, PD-1-binding antibody) in patients with locally advanced or metastatic liver cancer (NCT03680508).

### 5.2. Advantages of Immune Checkpoint Inhibitors in the Treatment of HBV-HCC

Immune checkpoint inhibitor therapy is more effective than traditional sorafenib therapy. It was found that HBV-positive patients were associated with a poor prognosis after receiving first-line conventional therapy with sorafenib, while patients with a low HBV load (≥10^4^ copies/mL of HBV DNA) had a significantly higher prognosis than patients with a high load (≥10^4^ copies/mL of HBV DNA) with HCC [111], suggesting that patients with HBV-HCC have a limited response to sorafenib treatment [112]. Atezolizumab, in combination with bevacizumab, maintained a clinically significant survival benefit compared with sorafenib in the IMbrave150 trial [113,114]. In addition, the efficacy of immune checkpoint inhibitors against HCC varies with different etiological factors. The first-line combination of Altizolizumab and bevacizumab significantly improved the prognosis of HCC patients with viral hepatitis, whereas it did not achieve significant efficacy in HCC patients with non-viral hepatitis. Therefore, HBV-HCC may be more suitable for immune checkpoint inhibitor therapy than HCC caused by other factors [113].

Compared with adoptive cell immunotherapy, immune checkpoint inhibitors have significantly higher efficacy and better prognosis in the treatment of HBV-HCC, although adoptive cell immunotherapies, such as CAR-T cell therapy and TCR-T cell immunotherapy, have achieved remarkable results in hematological malignancies. Due to the lack of tumor-specific antigens in solid tumors including HCC, the inhibitory effect of the tumor microenvironment on the activity of CAR-T cells, and the homing barrier of CAR-T cells, the efficacy of CAR-T cell immunotherapy in solid tumors is limited [115]. Meanwhile, for TCR-T cell immunotherapy, it is difficult to construct a population of TCR-T cells that can identify a reliable target with sufficient affinity and function, and it is associated with the effects of tumor antigen inhibition and the possible safety issues arising from the gene transfer of TCRs [116]. Hence, the efficacy of this strategy in solid tumors is not significant, and its application in clinical HCC treatment is still in a dilemma to a certain extent. In addition, tumor vaccines appear to be more challenging to treat HCC than immune checkpoint inhibitors. Although vaccines have long been considered a standalone treatment, they are more suitable for use in combination with immune checkpoint inhibitors or adoptive cell immunotherapy. HCC peptide vaccines with defined antigens usually target tumor-associated antigens; however, only a few strategies targeting tumor-associated antigens have reached the clinical stage, such as glycan-3 and telomerase [117]. Meanwhile, none currently in clinical stage have clinically significant results for drug development. Therefore, we believe that immune checkpoint inhibitors have irreplaceable advantages in the treatment of HBV-HCC.

### 5.3. Efficacy of Immune Checkpoint Inhibitors in the Treatment of HCC with Different Etiologies

In recent years, more and more researchers have been focusing on efficacy differences in immune checkpoint inhibitors in the treatment of HCC with various etiologies, and different opinions are held.

Some researchers consider that etiology is not a factor that affects the efficacy of immune checkpoint inhibitors. A study of the efficacy of viral status in immunosuppressive therapy for HCC revealed no significant difference in the objective response rate (ORR) between viral and non-viral HCC patients treated with immune checkpoint inhibitors; in any case, no difference was observed in ORR between HBV-HCC and HCV-HCC patients either [118]. However, Pfister D et al., by comparing with viral HCC, found that non-viral HCC, especially NASH-HCC, responded poorly to immune checkpoint inhibitor therapy. After anti-PD1 immune checkpoint inhibitor treatment, instead of regression, liver tumors showed increased fibrosis, while other non-NSAH responded to anti-PD1 treatment and were not associated with damage [119]. This suggests that, in NASH-HCC, the immune surveillance is impaired and CD8+ T cells contribute to HCC induction. Surprisingly, Murai H et al. seemed to draw a conclusion contrary to that of Pfister D et al. They examined the TME of ASH-HCC and NASH-HCC and found that the TME of steatosis HCC was characterized by immune enrichment, but immune depletion and high expression of PD-L1 [120]. After anti-PD-L1 treatment, the progression-free survival of steatosis HCC patients is significantly longer than that of non-steatosis HCC patients, so steatosis HCC patients are more likely to benefit from altezolizumab (anti-PD-L1) combined with bevacizumab and immunotherapy.

In agreement, several clinical trials support the finding of Murai H et al. In the ORR subgroup analysis of the IMbrave150 study, it was found that anti-PD-L1/VEGF had a good effect on non-viral HCC [114], and regardless of the cause, the CheckMate-040 study also showed that anti-PD-L1/anti-CTLA-4 treatment was effective [121]. Therefore, more evidence is needed to explain the effect of NASH etiology on immunotherapy in HCC patients. In summary, we agree with the notion that HCC with various etiologies may show different therapeutic efficacies when undergoing immune checkpoint inhibitor treatments; moreover, the combination of immune checkpoint inhibitors in the treatment of HBV-HCC might exert reliable effects beyond our expectation.

## 6. Potential Immune Checkpoint Therapeutic Targets

Although the immune checkpoint inhibitor treatment of HBV-HCC has achieved promising clinical therapeutic effects, there are still some checkpoint inhibitors that show acquired resistance during treatment [122]. In this context, new immune checkpoints are constantly required. Next, we introduce a few of the most recently discovered potential targets in HBV-HCC and the related proteins of these targets.

### 6.1. HHLA2

As a member of the immune checkpoint B7 family, HHLA2 (B7H7) is expressed on the surfaces of human monocytes, and also on the surfaces of B cells after stimulation by INF-γ [123]. HHLA2 is mainly present in cancer cells, such as liver cancer, breast cancer, lung cancer, and osteosarcoma [124]. In a recent study, HHLA2 was found not to be sufficient to fully activate T cells after binding to T cell surface receptors, but in the presence of both T cell surface receptors and CD28 signaling pathways, HHLA2 mainly plays an inhibitory role in T cell proliferation and cytokine activity [125]. HHLA2 binds to its ligand TMGD2 (CD28H) in HCC to activate the JAT/STAT signaling pathway, leading to the dysfunction of T cells and promoting the immune tolerance of hepatoma cells [126]. Clinical trials have shown that the expression of HHLA2 in hepatoma cells is closely related to the presence of CD8+ T cells in tumors and predicts good prognosis, suggesting the potential of HHLA2 as a therapeutic target for HCC treatment [127].

### 6.2. CCR4

CC chemokine receptor 4 (CCR4), with two ligands, CCL17 and CCL22, is a key chemokine receptor selectively expressed on normal T cells that mediates the entry of regulatory T cells into the tumor microenvironment [128], and it is mainly expressed in tumor-infiltrated Treg cells. By upregulating the expression of PD-1 and CTLA-4, CCR4 can stimulate the further secretion of more immunosuppressive factors, such as IL10 and IL-35, so CCR4 could enhance the effectiveness of immune checkpoint occlusion therapy [129]. In addition, Mogamulizumab is an inhibitor of CCR4 and has a favorable effect in the treatment of solid tumors with Treg cell depletion [130]. In HCC, increased CCR4 expression promotes the malignant proliferation of HCC and stimulates the migration and invasion of HCC through ERK/AKT/MMP2 signaling, so interfering with CCR4 may present an effective strategy for HCC treatment [129].

### 6.3. DDRs

Discoidin Domain Receptors (DDRs), including DDR1 and DDR2, are collagen receptors with tyrosine kinase activity that are closely associated with fibrosis and tumorigenesis [131]. As reported, the DDR1 extracellular domain contributes significantly to immune exclusion, providing great obstacles for immune cells’ infiltration into cancer tissue; thus, targeting DDR1 via monoclonal antibody could reverse this situation and provide a strategy for tumor treatment [132,133]. Meanwhile, DDR2 could be an effective target for enhancing tumor responses to anti-PD-1 according to recent research, which has proven that targeting DDR2 alongside PD-1 inhibitors exerts a more significant treatment effect among various cancers [134].

### 6.4. IL-27/IL-27R

As a powerful immunosuppressive cytokine, IL-27 inhibits the production of pro-inflammatory cytokines and affects the phenotype of CD4^+^ T cells. It has been shown that IL-27 and its receptors modulate various acute and chronic inflammatory diseases [135,136]. As imperative immune cells in the body, NK cells play an important role in killing aging and damaged cells, as well as in anti-tumor and immune regulation [137]. Recently, it has been reported that the IL-27 receptor can regulate the aggregation, activation, and cytotoxicity of NK cells, thus inhibiting HCC, which provides a new target for the treatment of hepatocellular carcinoma [138].

## 7. Conclusions

Given the complex immune microenvironment, immune checkpoint inhibitors in HCC enhance the immune response by reactivating immune cells and breaking immune tolerance, offering promising therapeutic strategies for HCC patients. These inhibitors block immune evasion by targeting corresponding immune checkpoints. For example, immune checkpoint inhibitors targeting CTLA-4 and PD-1/PD-L1 have revolutionized cancer treatment by inducing and restoring anti-tumor immunity. Therefore, tumor immunotherapy has undoubtedly become one of the most important methods of oncology treatment. In recent years, immune checkpoint therapy has developed rapidly, and many new immune checkpoints, such as LAG-3, TIGIT, and TIM-3, have been discovered; the corresponding inhibitors are now in clinical application or in clinical trials. As a new type of immunotherapy, immune checkpoints have opened another avenue for the treatment of patients with HCC, but are not limited to HCC and the efficacy of immune checkpoint inhibitor in treating HCC with different etiologies is worthy of further study.

HBV-HCC immune checkpoint therapy strategies include immune checkpoint monotherapy and immune checkpoint inhibitors combined with targeted drug therapy. Nivolumab, pembrolizumab, pembrolizumab, and tislelizumab are currently in clinical trials to evaluate outcomes in patients with advanced HCC treated with a monotherapy of immune checkpoint inhibitors. Due to the low overall response rate to immunotherapy, fewer patients with advanced HBV-HCC can benefit. In addition, despite targeting immune checkpoints, inhibitors will inevitably cause side effects in important organs in patients due to dose tolerance, individual physical differences, and other conditions [139,140]. Therefore, immunotherapy combined with other treatment methods may provide a promising direction in the future application of immunotherapy checkpoint inhibitors. Combination therapies include ICIs combined with other ICIs, combined tyrosine kinase inhibitors (TKIs), anti-vascular endothelial growth factor antibodies (anti-VEGF), and other drugs. Many clinical trials are currently evaluating the potential of ICIs in combination with advanced HCC. Furthermore, oncolytic viruses are a class of natural or recombinant viruses that can selectively infect and kill tumor cells without damaging normal cells. They can improve the clinical survival benefits of patients with HCC and improve the survival period and objective effective rate of patients with advanced HCC. Therefore, oncolytic therapy combined with immune checkpoint blocking therapy is a new concept for the treatment of hepatocellular carcinoma [141]. However, there are still many unresolved concerns regarding combination therapy, including safety, drug resistance, toxic side effects, and the optimal combination approach and dose for application.

In the future, further evaluations of the efficacy and mechanisms of action of immune checkpoint inhibitors are required to provide better treatment strategies for HCC patients.

## Figures and Tables

**Figure 1 vaccines-11-00614-f001:**
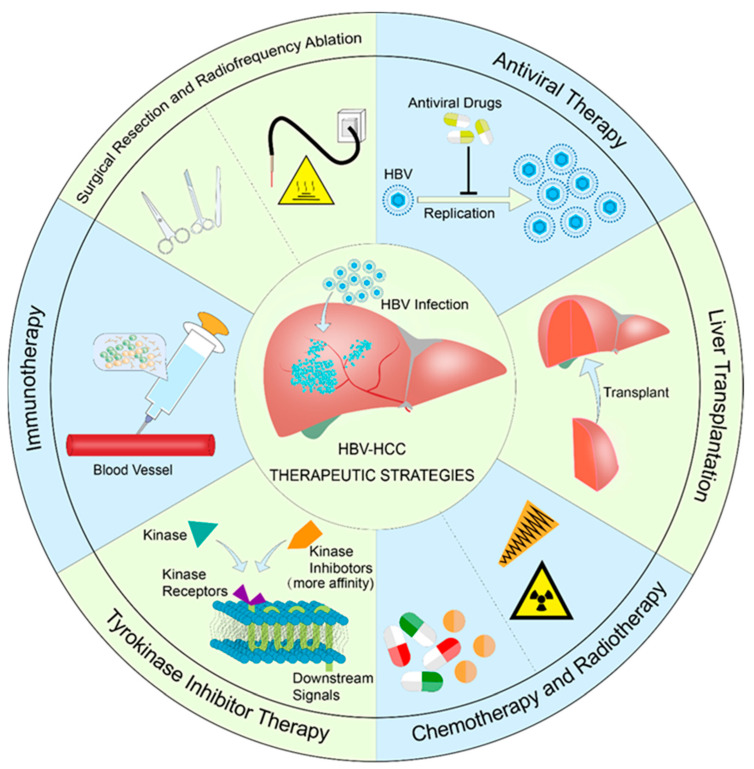
HBV-HCC therapeutic strategies.

**Figure 2 vaccines-11-00614-f002:**
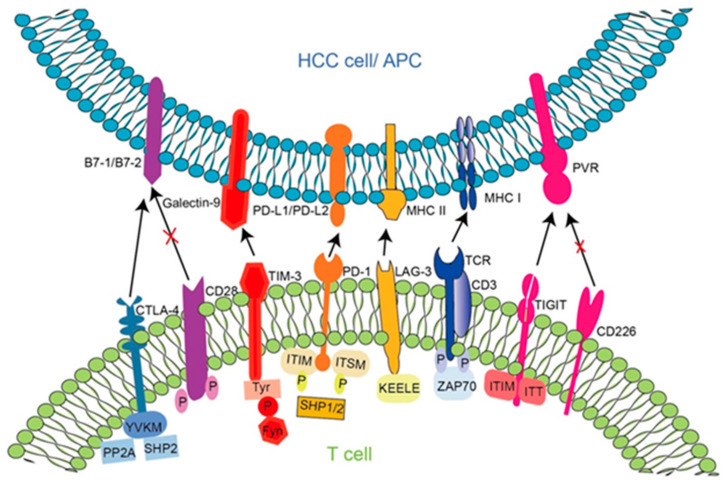
Immune checkpoint molecules and their ligands.

**Table 1 vaccines-11-00614-t001:** HBV genotypes and the oncogenic mechanisms of HBV-HCC.

HBV Genotype	A	B	C	D	E	F	G	H	I	J (Putative)
Occurrence and development of HBV-HCC	Direct oncogenic mechanism of HBV-HCC			Insertion mutations						
				HBV causes genomic instability						
				Prolonged expression of viral proteins affects cell function						
	Indirect oncogenic mechanisms of HBV-HCC			Immune-mediated indirect oncogenic mechanisms of HBV						

**Table 2 vaccines-11-00614-t002:** Summary of clinical trials of immune checkpoint inhibitors in HBV-HCC.

Immune Checkpoint	Current FDA-Approved Inhibitors	No. of Ongoing Clinical Trials Related to HCC *	Hyperlinks and References
PD-1	Nibolumab (FDA-approved for the treatment of HCC following sorafenib)	88	ClinicalTrials.gov [68,69]
	Pembrolizumab (FDA-approved for the treatment of HCC following sorafenib)	76	ClinicalTrials.gov [70]
	Cemiplimab (FDA has not approved it for HCC treatment)	4	ClinicalTrials.gov
	Dostarlimab (FDA-approved for treatment of unresectable liver cancer)	1	ClinicalTrials.gov
CTLA4	Tremelimumab (FDA-approved for treatment of unresectable liver cancer)	20	ClinicalTrials.gov [71,72]
	Ipilimumab (FDA-approved for the treatment of hepatocellular carcinoma following sorafenib)	22	ClinicalTrials.gov [69]
LAG-3	Relatlimab (FDA has not approved it for HCC treatment)	4	ClinicalTrials.gov [73]
TIGIT	Tiragolumab has been granted breakthrough drug status by the FDA, and Vibostolimab has initiated clinical trials related to HCC	2	ClinicalTrials.gov1 ClinicalTrials.gov2
TIM-3	Cobolimab is undergoing a series of clinical trials	1	ClinicalTrials.gov

* Clinical trial numbers are based on search term inputs; thus, some results include clinical trials involving HCC and ICP terms, and the actual number might vary.

## Data Availability

Not applicable.

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
