# Peer review of "Immune Checkpoint Inhibitors in HBV-Caused Hepatocellular Carcinoma Therapy"

_vaccines, 2023, doi:10.3390/vaccines11030614_

Round 1
Reviewer 1 Report
1. This review article summarizes current knowledge about anti-cancer therapies using immune checkpoint inhibitors (ICIs) against HBV-HCC. Overall, there was no really new or novel insight into how ICI therapies are more favorable than other approaches against HBV-HCC but simply updating and summarizing what has been known about ICI therapies for HBV-HCC in the literature (abundant papers doing the same thing).
2. Why did they not make ICI therapies as a general anti-HCC strategy and only specifically emphasize HBV-HCC?
3. The authors should have addressed more why ICI therapies are particularly more favorable for HBV-HCC. Are they not suitable for HCV-HCC or other types of HCC?
4. Should different ICI therapies be applied to different types of HCC due to different immune checkpoint marker expressions on different types of HCC cells and on their immune cell surfaces?
5. They should also provide updated and solid evidence from references to support their thinking.
Author Response
Response to Reviewer1 Comments
Dear reviewer,
We appreciate your constructive comments and suggestions, which have greatly helped us to improve our manuscript. Our point-by-point responses to each of the comments of the reviewers are attached in the following part.
Point 1: This review article summarizes current knowledge about anti-cancer therapies using immune checkpoint inhibitors (ICIs) against HBV-HCC. Overall, there was no really new or novel insight into how ICI therapies are more favorable than other approaches against HBV-HCC but simply updating and summarizing what has been known about ICI therapies for HBV-HCC in the literature (abundant papers doing the same thing).
Response 1: We thank the reviewer for pointing this out for us. It is possible that the novelty of this review is still not satisfactory, while the content is comparatively systematic, aiming to introduce and update our knowledge on HBV-HCC at the present time, including molecule characterization, typing, occurrence, therapeutic strategies, and ICI therapies. Certainly, the ICI therapies have already been reviewed in other articles, so this review selectively focused on the field of HBV-HCC; thus, the immune checkpoints and inhibitors are all considered in this context of HBV-HCC, and clinical trials were also reviewed in terms of HBV-HCC.
Point 2: Why did they not make ICI therapies as a general anti-HCC strategy and only specifically emphasize HBV-HCC?
Response 2: HBV is a major cause of HCC, and HBV-HCC is also very prevalent and a severe health issue in the East Asian region. Our review is mainly focused around this point and discusses it from different angles. If we pay attention to ICI therapies as anti-HCC and not HBV-HCC, this would be too extensive to illustrate in detail, and other aspects of HBV-HCC would be difficult to include, which would complicate the aim of our work.
Point 3: The authors should have addressed more why ICI therapies are particularly more favorable for HBV-HCC. Are they not suitable for HCV-HCC or other types of HCC?
Response 3: ICI therapies are also suitable for other types of HCC, not limited to HBV-HCC. While the treatment effect shows differences in different types of HCC, ICI therapies provide a better response to viral HCC compared to non-viral HCC, which has not been mentioned, and we try to complement this point.
Point 4:Should different ICI therapies be applied to different types of HCC due to different immune checkpoint marker expressions on different types of HCC cells and on their immune cell surfaces?
Response 4: Different ICI therapies are also suitable for different types of HCC, but with different treatment effects. In addition, differences in immune checkpoint expression existing in different types of HCC cells’ and immune cells’ surfaces will influence the selection of ICI therapies, but the expression of immune checkpoints will vary according to individual patients, the course of disease and other factors, so the type of ICI therapy to be be selected will be determined by the clinical condition of each patient.
Point 5: They should also provide updated and solid evidence from references to support their thinking.
Response 5: Thanks for your suggestions; we have made some modifications in the revision.
Reviewer 2 Report
The authors reviewed and discussed current status and future progress of immuno-check point inhibitors in HBV associated HCC.
1. How about no-HBV associated HCC? Does the immuno-check point inhibitor works the same?
2. The authors should make more statement of current treatment on HCC, especially in the section of 4.5.
Author Response
Response to Reviewer2 Comments
Dear reviewer,
We appreciate your constructive comments and suggestions, which have greatly helped us to improve our manuscript. Our point-by-point responses to each of the comments of the reviewers are attached in the following part.
The authors reviewed and discussed current status and future progress of immuno-check point inhibitors in HBV associated HCC.
Point 1:How about no-HBV associated HCC? Does the immuno-check point inhibitor works the same?
Response 1: The pathogenesis mechanisms of various types of HCC are different, and the treatment strategies will also present differences. According to recent research, viral HCC will respond better to immune checkpoint therapy than non-viral HCC.
Point 2: The authors should make more statement of current treatment on HCC, especially in the section of 4.5
Response 2: Thanks for your suggestion; we have considered the current treatment of HCC in the revision.
Reviewer 3 Report
Authors have attempted a comprehensive view of immune checkpoint inhibitors in virus mediated HCC.
They are recommended to include a table for section 2 and 3 of the review for quick access to information.
Also, please proofread the manuscript for language and grammar. The language in all sections has convoluted sentence construction hence, it is often difficult to follow the scientific idea.
The conclusion section requires a more detailed view of future perspectives in the field.
Author Response
Response to Reviewer3 Comments
Dear reviewer,
We appreciate your constructive comments and suggestions, which have greatly helped us to improve our manuscript. Our point-by-point responses to each of the comments of the reviewers are attached in the following part.
Point 1: They are recommended to include a table for section 2 and 3 of the review for quick access to information.
Response 1: Thanks for the suggestions. We have already added a table for Parts 2 and 3 for easier access to information.
Point 2: Also, please proofread the manuscript for language and grammar. The language in all sections has convoluted sentence construction hence, it is often difficult to follow the scientific idea.
Response 2: We thank the reviewer for pointing this out for us. We have reedited the language and grammar of the manuscript to make it easier to follow the scientific ideas.
Point 3: The conclusion section requires a more detailed view of future perspectives in the field.
Response 3: Following the reviewer’s suggestion, we have included the future prospects of immune checkpoint inhibitors in the treatment of hepatocellular carcinoma caused by HBV in more detail in the Conclusions.
Round 2
Reviewer 1 Report
Merely introducing and updating the knowledge on HBV-HCC, including molecule characterization, typing, occurrence, therapeutic strategies, and ICI therapies is not novel and insightful enough. The authors are suggested to add more discussions in the text.
At least, the authors should use a section to deeply discuss why they emphasized immune checkpoint inhibitors, among all other possible therapeutics, in HBV-HCC therapy, meaning why ICI therapies are more favorable than others.
In addition, they should also carefully discuss the epidemiology of HBV-HCC by comparing it to other types of HCC to reason why HBV-HCC should particularly be focused.
Lastly, they should point out what this review article could bring that is something new or novel compared to the abundant references related to the same issues.
Author Response
Dear reviewer,
We appreciate your constructive comments and suggestions, which have greatly helped us to improve our manuscript. Our point-by-point responses to each of the comments of the reviewers are attached in the following.
Point 1: At least, the authors should use a section to deeply discuss why they emphasized immune checkpoint inhibitors, among all other possible therapeutics, in HBV-HCC therapy, meaning why ICI therapies are more favorable than others.
Response: Thank you very much for your advice and we have made changes accordingly as you suggested. We added the ”5.2 Advantages of immune checkpoint inhibitors in the treatment of HBV-HCC” section (Lines 494-528) to discuss the advantages of ICIs in the treatment of HBV-HCC thoroughly to clarify why ICIs therapy should be emphasized.
Point 2: In addition, they should also carefully discuss the epidemiology of HBV-HCC by comparing it to other types of HCC to reason why HBV-HCC should particularly be focused.
Response: We agree with the review on this. In the revision, we added the epidemiology of HBV-HCC in the introduction (Lines 42-62)
Point 3: Lastly, they should point out what this review article could bring that is something new or novel compared to the abundant references related to the same issues.
Response: This is great suggestion and this part is also the essence of this review, despite that we know our views may not be fully satisfied by the readership. In the revision, we discussed that the advantages of immune checkpoint inhibitors in treating HBV-HCC and whether there are differences in the efficacy of immune checkpoint inhibitors in the treatment of HCC of various etiologies, which are two major concerns worth exploring, as stated in "5.3 Efficacy of immune checkpoint inhibitors in the treatment of HCC with different etiologies." section (Lines 529-560).
In addition, we also made minor revisions to the abstract and conclusion for the consistency of the manuscript.